# OpenReview forum: "Beyond Forgetting: Representation Misdirection Elicits Controllable Side Behaviors and Capabilities"
_TMLR — Under review for TMLR_

### Review · Reviewer_8MtM · 2026-06-22

**Summary Of Contributions:**

Summary of Contributions:
The paper explores the mechanics of Representation Misdirection (RM) in Large Language Model (LLM) unlearning by viewing it through the lens of the Linear Representation Hypothesis. The authors demonstrate that instead of merely injecting random noise to make a model "forget" specific data, researchers can steer the model's internal representations toward specific, high-level concept vectors.

Key Strengths:
1. Strong Theoretical Grounding: The paper bridges empirical unlearning observations with theoretical foundations, providing mathematical proofs supporting both additive and ablative interventions based on the Linear Representation Hypothesis.
2. Comprehensive Evaluation: The authors rigorously validate their hypothesis across a diverse set of tasks and multiple models (e.g., Zephyr-7B, Mistral-7B, and Llama-3-8B). The experiments convincingly demonstrate behavioral control across domains like sentiment, language, and refusal.

Key Weaknesses
1. High Vulnerability to Knowledge Recovery: Extensive robustness testing in the appendix reveals that the unlearned models are easily compromised. White-box attacks (specifically fine-tuning, orthogonalization, and pruning) successfully restore the model's supposedly forgotten knowledge.
2. Limitations with Complex Reasoning: The RAd method struggles to elicit stronger capabilities on complex mathematical reasoning tasks, such as GSM8K. The authors attribute this to the inherent difficulty of compressing a complex, multi-step reasoning process into a single one-dimensional vector.
3. Output Quality Trade-offs: The paper identifies a distinct trade-off when using concept directions. Because the concept vector may become entangled with existing concepts in the retain domain, it can degrade the grammatical correctness and coherence of the model's outputs on general (retain) tasks.
4. Restricted Model Scale: The experiments were conducted entirely on 7-8B parameter models due to computational constraints. The authors note that restricting updates to a subset of components in smaller models risks missing phenomena that might manifest differently in larger, state-of-the-art models.

**Audience:**

Yes

**Audience Explanation:**

Evaluation: Yes, the findings are highly relevant to TMLR's audience.

1. Intersection of Unlearning and Alignment: This paper provides a novel perspective by connecting Representation Misdirection (RM) with behavioral alignment, demonstrating that steering forget-representations can simultaneously control traits like truthfulness, sentiment, and refusal.
2. Novel Mechanistic Insights: Researchers interested in the "black box" nature of LLMs will value the paper's application of the Linear Representation Hypothesis.
3. Counterintuitive Capability Enhancements: Typically, unlearning is viewed strictly as a mechanism for capability reduction or knowledge removal. The authors demonstrate that Representation Addition (RAd) can actually enhance side capabilities, such as in-context learning (ICL) on linguistic and knowledge tasks. This dual-purpose mechanism is an intriguing finding for practitioners looking to optimize model performance while removing harmful data.

**Broader Impact Concerns:**

The Broader Impact Statement needs to be expanded. The authors must explicitly acknowledge the dual-use risks of their methodology: specifically, that the exact same mechanisms used for unlearning and capability enhancement can be trivially inverted to execute targeted safety ablations and degrade model truthfulness.

**Claims And Evidence:**

Yes

**Claims Explanation:**

Evaluation: Yes, the claims are supported by accurate, convincing, and clear evidence.

- The authors do not merely rely on empirical trial-and-error; they provide a clear mathematical foundation for their methodology. By leveraging the established Linear Representation Hypothesis, they formulate precise definitions for Representational Addition (RAd) and Representational Ablation (RAb).
- Broad and Diverse Empirical Validation: The paper provides a highly convincing array of experiments across multiple distinct domains to validate the Controllable Emergent Capability Hypothesis. Evidence is clear and consistently structured across two main pillars:  Behavioral Control and In-Context Learning (ICL) Enhancement.
- Rigorous Experimental Controls: The evidence is highly convincing because of the extensive control variables and baselines used: 1) Concept vs. Random Vector Baselines, 2) Comparison to State-of-the-Art (SOTA).

**Requested Changes:**

Critical for Acceptance:
1. Qualitative Examples of Retain-Set Degradation: Section 5.3 identifies a critical trade-off; while RAd with concept directions improves output coherence on the forget set, it degrades the quality of outputs on the retain set (MMLU) when compared to random vectors. Relying solely on an LLM-as-a-judge win-rate graph obscures the practical nature of this degradation. Provide explicit, qualitative examples of generated text where the retain-set output degraded. Show the reader exactly what this "representational shift" looks like in practice (e.g., does the model output gibberish, or does it inappropriately apply the injected concept to general queries?).
2. Deeper Investigation into Reasoning Failures: The paper notes that RAd with a Chain-of-Thought (CoT) direction showed limited effectiveness on complex reasoning tasks like GSM8K. The authors hypothesize this is due to the 7B model's capacity or the difficulty of representing a long CoT prompt as a one-dimensional vector. Because this represents a significant boundary to the paper's core hypothesis, it requires further exploration. Conduct a targeted ablation testing whether extracting concept vectors from intermediate reasoning steps (rather than "Let's think step by step" prompt) yields better linear representations for RAd to clarify if this is a fundamental limitation of the method or merely an issue with prompt selection.

Would Strengthen the Work:
1. Experiments on Larger Scale Models: The current empirical validation is restricted to 7B and 8B parameter models due to stated computational constraints. Because feature entanglement in the latent space can behave differently as parameter counts scale up, the generalizability of the findings is slightly constrained. If compute allows, test the RAd/RAb methodology on at least one larger model (e.g., a 30B+ or 70B parameter model) on a subset of the tasks to verify that the Controllable Emergent Capability Hypothesis scales consistently.
2. Comprehensive Layer-wise Analysis: Appendix G.4 provides a brief ablation showing that the controllability phenomenon persists at layer 15 but carries a higher risk of damaging general performance due to feature entanglement. Expand this into a layer-wise heatmap or a broader summary chart showing RAd/RAb effectiveness across early, middle, and late layers.

---

> ### Author Response · Authors · 2026-07-06
>
> We thank the Reviewer for your time and effort in reviewing our paper. Below, we will keep the response to your request changes (RC) brief and focused. The newly added content in the revision is highlighted in blue text.
>
> ### Critical for Acceptance:
> **RC1**
> > Qualitative Examples of Retain-Set Degradation: Section 5.3 identifies a critical trade-off; while RAd with concept directions improves output coherence on the forget set, it degrades the quality of outputs on the retain set (MMLU) when compared to random vectors. Relying solely on an LLM-as-a-judge win-rate graph obscures the practical nature of this degradation. Provide explicit, qualitative examples of generated text where the retain-set output degraded. Show the reader exactly what this "representational shift" looks like in practice (e.g., does the model output gibberish, or does it inappropriately apply the injected concept to general queries?).
>
> **Answer:**
> + We thank the Reviewer for this suggestion. We have added Appendix G.7, which provides qualitative examples of generated outputs on both retain queries (MMLU) and forget queries (WMDP).
> + Furthermore, we conducted an additional analysis of the latent representations in Section 5.1. By comparing the distributions of the inner products between latent representations and the target concept vector, we show that RAd shifts the representations of both forget and retain samples toward the target concept, while RAb suppresses the corresponding concept component.
>
> **RC2**
> > Deeper Investigation into Reasoning Failures: The paper notes that RAd with a Chain-of-Thought (CoT) direction showed limited effectiveness on complex reasoning tasks like GSM8K. The authors hypothesize this is due to the 7B model's capacity or the difficulty of representing a long CoT prompt as a one-dimensional vector. Because this represents a significant boundary to the paper's core hypothesis, it requires further exploration. Conduct a targeted ablation testing whether extracting concept vectors from intermediate reasoning steps (rather than "Let's think step by step" prompt) yields better linear representations for RAd to clarify if this is a fundamental limitation of the method or merely an issue with prompt selection.
>
> **Answer:**
> + *On reasoning capability of the base model*: We hypothesized that the limited effectiveness of RAd and RAb on Zephyr-7B and Mistral-7B is primarily due to their weak reasoning capability, rather than a falsification of the proposed hypothesis. To validate this, we conducted additional experiments on Qwen2.5-7B, a stronger reasoning model. The new results (Tables 8 and 9, in the revision) show that both RAd and RAb become significantly more effective on GSM8K and GSM-Plus, supporting our hypothesis that the effectiveness depends on the underlying reasoning capability of the base model.
> + *More complex template*:  To further investigate whether the observed limitation stems from the choice of prompt, we constructed a 5-shot context direction, which is longer and more complex than the standard zero-shot CoT prompt ("Let's think step by step."). The new results in Table 9 (in the revision) demonstrate that RAd and RAb remain effective when the concept direction is extracted from this longer context, particularly on Qwen2.5-7B.
>
> ### Would Strengthen the Work:
> > Experiments on Larger Scale Models: etc.
>
> > Comprehensive Layer-wise Analysis: etc.
>
> **Answer:**
> 1. *Experiments on larger models.* Due to computational constraints, we were unable to perform experiments on 30B+ or 70B models in this revision. We acknowledge this as a limitation.
> 2. *Layer-wise analysis*. We expanded our layer analysis by conducting additional experiments with RAd and RAb at a deeper layer (layer 30) on TruthfulQA (both open-ended and multiple-choice), in addition to the previously reported layers (layer 7 and 15). The new results are presented in Table 20. They consistently support our hypothesis that concept representations become increasingly entangled with other semantic features in deeper layers. Consequently, unlearning with extracted concept directions from deeper layers increases the risk of degrading the unlearned model's general capabilities. **Although we were unable to perform a full layer-by-layer heatmap due to computational constraints**, we believe these additional experiments provide stronger empirical evidence for our claim of feature entanglement across network depth.
>
> > Broader Impact Concerns:
>
> **Answer:** We have updated the broader impact statement to acknowledge the dual-use risks of our methods.
>
> **We thank you for helping improve our work. We are committed to addressing remaining concerns in the final revision.**

---

### Review · Reviewer_CUvB · 2026-06-22

**Summary Of Contributions:**

The paper investigates Representation Misdirection (RM) in Large Language Model (LLM) unlearning through the lens of the Linear Representation Hypothesis (LRH). Standard RM approaches (like RMU) typically redirect latent representations of forget-items toward a fixed random vector to induce forgetting. The authors argue that this introduces uncontrolled noise and negative side effects.

As an alternative, they propose two conceptual frameworks:
- Representational Addition (RAd): Steering the forget-representations toward a specific, known concept vector.
- Representational Ablation (RAb): Projecting forget-representations onto the null space of a specific concept vector.

The core claim is that by substituting random target vectors with concept-specific vectors, the unlearning process can simultaneously elicit predictable, controllable side behaviors (e.g., truthfulness, sentiment, language) or enhance specific capabilities (e.g., in-context learning) matching that concept, all while executing the unlearning task. The authors evaluate their approach on the WMDP (Biology/Cyber) datasets using Zephyr-7B, Mistral-7B, and Llama-3-8B.

The paper is structured in three parts: Section 3 introduces a conceptual and theoretical analysis of the existing MR unlearning strategy, with the definition of the RAd and RAb methods; Section 4, conducts an experimental analysis of the proposed unlearning method; and Section 5 complements the empirical results with some ablations of the method.

  Hereby, I list a summary of the main claims (Hx):

(H1) RMU (with random directions) induces undesirable and uncontrolled side effects.

(H2) RM with concept-related directions (RAd or RAb) can be used for unlearning

(H3)  RM with concept-related directions (RAd or RAb) induces predictable side effects

**Additional Comments:**

Minor Points & Clarifications

- Terminology: Phrases like "forget-representations" and "forget-input" are jargon-heavy and reduce readability in the abstract and introduction. Standardizing to "latent representations of the forget set" would be much clearer.
- Incomplete Baselines: In Table 10 (Comparison to current unlearning methods), the authors only provide results for Zephyr-7B. To support their claims robustly, Mistral-7B should be included in this cross-method comparison as well.
- Statistical Rigor: The tables lack error bars or confidence intervals, making it difficult to judge if minor fluctuations (e.g., random RAb slightly increasing metrics in Table 1) are statistically significant or just noise.

**Audience:**

Yes

**Audience Explanation:**

Yes. Machine unlearning is a critical subfield of LLM safety and alignment, and mitigating its collateral damage (specifically, the degradation of a model's general capabilities) is a highly active area of research. Using the Linear Representation Hypothesis (LRH) to diagnose and steer these side effects offers a mechanistically grounded perspective. This approach will be of significant interest to researchers working at the intersection of mechanistic interpretability, representation engineering, and model editing.

**Broader Impact Concerns:**

The paper should include a Broader Impact Statement addressing the potential dual-use risks inherent to the proposed methodology.

**Claims And Evidence:**

No

**Claims Explanation:**

1) The authors claim (H1) that standard RM with random directions induces catastrophic, uncontrolled side effects (such as incoherent or gibberish text). However, **they provide no concrete empirical baseline measurements or literature references in the introduction to substantiate how severe this "gibberish" effect actually is in practice**. Crucially, their text quality analysis in Section 5.3 shows that while concept vectors improve generation quality on the forget set, they actually degrade text quality on the retain set (MMLU) compared to random vectors. This undercuts the claim that concept vectors are a strictly superior, safer alternative.

2) **The theoretical analysis in Section 3.2 essentially restates the definitions of the LRH from prior work.** Assuming the LRH is true, it is mathematically trivial that a linear shift in the representation space will linearly shift the log-odds of the corresponding concept output.
- The theory fails to address the actual challenge of the paper: learnability via optimization.
- The proposed methods change the weights of the model via fine-tuning (Eq. 10 and 11) to enforce this representation shift on the forget set. The theory assumes a static representation space and does not explain how the model's overall weights alter during optimization, or why changing weights based only on the forget-set distributions causes specific generalized capabilities to emerge.

3) **The empirical results does not fully demonstrate the utility (H2) and reduced side-effect (H3) of concept vectors for unlearning**:
- Performance Trade-off: In Table 1, standard RM with a random vector (RAd w/ random) performs on par with or noticeably better than the concept variant (RAd w/ truth) at dropping WMDP accuracy.
- Severe General Capability Collapse: RAd w/ truth causes a drop in general language modeling capability. This contradicts the claim of reduced side-effect, which should lead to a preservation of general capabilities. The authors note in Section 5.3 that the concept direction is likely entangled with retain domains, causing an unexpected representation shift. This implies that the side effects are not entirely controlled or predictable, weakening the evidence for H3.

4) Practical Motivation of Specific Tasks. While controlling sentiment or truthfulness during unlearning makes sense conceptually, other configurations lack practical utility. For instance, in Section 4.1.4, the authors show that unlearning hazardous biological data while forcing the model to reply in French (RAd w/ en -> fr) works. It is unclear why a practitioner would ever want an unlearning mechanism to trigger a random language shift as a side behavior. The paper treats this as a proof of behavioral control, but as an unlearning utility, it is poorly motivated.

**Requested Changes:**

To meet TMLR’s criteria for publication, the authors must address the gap between their conceptual claims and the provided empirical evidence. The following changes are requested:

1. Empirical Validation of Side-Effect Reduction (H1)

**Quantify the Baseline**: Provide concrete baseline measurements or explicit citations in the introduction to substantiate the claim that standard RM causes "catastrophic, uncontrolled side effects" (like gibberish text).
**Prove Superiority**: Address the contradiction in Section 5.3 where RAd w/ truth causes a worse drop in general knowledge (MMLU) than the random vector baseline. The authors must demonstrate via additional experiments or deeper analysis that their method actually reduces overall undesirable side effects compared to standard unlearning baselines.

2. Streamline Theory and Move to Appendix

**Condense Section 3.2**: The theoretical portion largely restates established properties of the LRH without explaining the optimization dynamics of the fine-tuning process. Reduce this section to its absolute core insights and move the extensive algebraic derivations to the Appendix to free up space for empirical analysis.

3. Expand Related Work

**Contextualize the Method:** Integrate a more robust literature review into the main text. Specifically, position this work alongside broader representation engineering frameworks and alternative unlearning paradigms (e.g., Gradient Difference, DPO-based unlearning, and targeted pruning) to clearly delineate the paper's novel contributions.

4. Improve Writing Quality and Reduce Jargon

**Enhance Readability**: The manuscript is currently difficult to digest on a first reading. Avoid non-standard jargon like "forget-representations" or "forget-input." Instead, use precise terminology (e.g., "latent representations of the forget-set distributions") and take the time to carefully introduce and define each core concept when it first appears.

5. Consolidate Experimental Results

Rather than treating each concept (sentiment, truthfulness, language, ICL) in isolation across separate subsections and individual tables, consolidate them into a unified comparative table. This will make the experimental narrative much more direct and free up layout space for the missing baseline evaluations.

6. Increase Model Architecture Diversity

**Test Independent LLM Families**: While the paper currently lists three models, Zephyr-7B is a fine-tuned derivative of Mistral-7B, meaning the empirical evaluation relies heavily on only two distinct base architectures (Mistral and Llama-3). To make the claims robust and generalizable, the authors should evaluate their method on at least two additional, structurally independent model family.

---

> ### Author Response · Authors · 2026-07-06
>
> We thank the Reviewer for your time and effort in reviewing our paper. Below, we will keep the response to your request changes (RC) brief and focused. The newly added content in the revision is highlighted in blue text.
>
> ### Major Changes:
> **RC1**
> > 1. Empirical Validation of Side-Effect Reduction (H1)
>
> **Answer:** We would like to clarify that:
>
> (1) our intention **was not to claim that RMU with a random direction inherently induces undesirable or uncontrolled side effects**. Rather, this observation was reported in prior work by Shen et al. [1], who argued that RMU may lead to incoherent or gibberish outputs (c.f. Figure 1 in [1]) and proposed the use of reference prompts as target directions.
>
> (2) We acknowledge that the original text in the Introduction did not attribute this claim sufficiently clearly. To address this issue, we have revised the Introduction to explicitly attribute the concern about random direction to prior work and to clarify that **our contribution is instead to investigate the mechanistic role of the target direction in controlling unlearned models' behaviors and capabilities**.
>
> (3) We would also like to clarify the objective of RAd and RAb. These two methods are conceptual models introduced to empirically test Hypothesis 1 and to understand the mechanistic role of target directions in RM. They are not proposed as unlearning methods intended to improve unlearning performance or establish a new SOTA, as stated in both the Abstract and the Introduction.
>
> (4) **We acknowledge the limitation of RAd w/ concept direction that degrades generated text quality in retain queries. We explicitly discuss this limitation in Section 5.4**. Furthermore, in this revision, we include additional qualitative examples of model generations on both retain queries (MMLU) and forget queries (WMDP) in Appendix G.7, providing further insight into the trade-offs.
>
> **RC2-5**
> > 2. Streamline Theory and Move to Appendix, etc
> 3. Expand Related Work, etc
> 4. Improve Writing Quality and Reduce Jargon, etc
> 5. Consolidate Experimental Results, etc
>
> **Answer:**
> + **RC2**: We think this is a good suggestion. In the revised manuscript, we have moved the theoretical discussion in the main text to Appendix E.
> + **RC3**: We have expanded the related work in the main text. We position our work alongside broader RM methods and other unlearning paradigms, and deferred the full related work to Appendix A.
> + **RC4**: In the revision, we have reduced the use of non-standard terminology throughout the paper, replacing jargon with more standard terms.
> + **RC5**: We understand the concern, but we believe that it is hard to consolidate them into a unified comparative table/figure since different tasks require different evaluation protocols, metrics, and experimental settings.
>
> **RC6**
> > 6. Increase Model Architecture Diversity, etc
>
> **Answer:** To further ensure the claims' generalization, we conducted additional experiments of Qwen2.5-7B model on reasoning task in Sec. 4.2.2, Qwen2.5-7B and Phi-4-mini on TruthfulQA tasks in Appendix G.9.
>
> ### Major Changes:
>
> > Terminology: Phrases like "forget-representations" and "forget-input" are jargon-heavy and reduce readability in the abstract and introduction. Standardizing to "latent representations of the forget set" would be much clearer.
> Incomplete Baselines: In Tab. 10 (Comparison to current unlearning methods), the authors only provide results for Zephyr-7B. To support their claims robustly, Mistral-7B should be included in this cross-method comparison as well.
> Statistical Rigor: The tables lack error bars or confidence intervals, making it difficult to judge if minor fluctuations (e.g., random RAb slightly increasing metrics in Table 1) are statistically significant or just noise.
>
> **Answer:** In the revision, we have reduced the use of non-standard terms throughout the paper. We conducted additional experiments comparing RAd with current unlearning methods with the Mistral-7B in Figure 3 (Tab 10-> Fig 3)
>
> > Broader Impact Concerns
>
> **Answer:** We have updated the broader impact statement to acknowledge the dual-use risks of our methods.
>
> ### References
>
> [1] Shen, William, et al. "LLM unlearning via neural activation redirection." Advances in Neural Information Processing Systems 38 (2026): 44253-44290.
> **Answer:**
>
> **We thank you for helping improve our work. We are committed to addressing remaining concerns in the final revision.**

---

### Review · Reviewer_XYdh · 2026-06-23

**Summary Of Contributions:**

This paper studies representation misdirection (RM) methods for LLM unlearning, focusing on the role of the target vector used to redirect forget-set representations. The authors reinterpret RM through the linear representation hypothesis and propose two variants: Representational Addition (RAd), which pushes forget representations toward a concept direction, and Representational Ablation (RAb), which removes components along a concept direction. The paper argues that these operations do more than reduce performance on forget tasks: they can also induce or suppress side behaviors tied to the chosen concept.

Strengths:
1. The paper evaluates unlearning on WMDP and MUSE and studies several side behaviors and capabilities, including TruthfulQA truthfulness, SST2 sentiment, refusal behavior, language control, simple ICL tasks, and GSM-style reasoning. The bidirectional evidence from RAd and RAb is useful: for example, adding a truthfulness direction improves TruthfulQA metrics, while ablating it reduces them; adding a language direction increases target-language presence, while ablating it suppresses that language.

2. The paper also includes comparisons to several unlearning baselines and an informative robustness appendix showing that recovered knowledge remains a serious issue.

Weaknesses:
1. The title and framing are broader than what the evidence establishes. The paper studies a specific family of representation misdirection methods in unlearning settings, and its unlearning evidence is primarily benchmark-level performance suppression rather than a formal or empirical guarantee of durable erasure. Thus, the title-level claim that "Machine Unlearning Elicits Controllable Side Behaviors and Capabilities" overgeneralizes from concept-directed RM to machine unlearning more broadly. A more precise framing would be that representation misdirection for unlearning can elicit controllable side behaviors and capabilities.

2. The experiments lack a direct steering-only baseline using the same semantic concept vectors. The random-vector controls are useful and show that arbitrary misdirection does not induce the same side behaviors, but the paper does not rule out the possibility that truthfulness, sentiment, refusal, language, or ICL changes would also arise from ordinary concept-vector steering without any forget-set RM objective.

**Audience:**

Yes

**Audience Explanation:**

The paper should be of interest to researchers working on machine unlearning, representation engineering, activation steering, model editing, and LLM safety. It connects two active areas: unlearning by representation misdirection and concept-vector steering. The finding that an unlearning target vector can systematically induce side behaviors is useful both as a risk analysis and as a possible design principle.

For example, the refusal experiments show a concrete safety-relevant risk: a procedure intended for unlearning can either increase refusals on harmless prompts or reduce refusals on harmful prompts, depending on the direction used. Similarly, the language and sentiment experiments show that the target vector can shape outputs in predictable ways.

**Claims And Evidence:**

No

**Claims Explanation:**

Currently, these claims lack precise, convincing, and clear supporting evidence.

- The title and main framework are too broad and inconsistent with the evidence. The paper describes the phenomenon as caused by "machine forgetting," but the experiment investigates a specific representational misleading mechanism, not certified machine unlearning.

- The experiment fails to distinguish between RM-based forgetting and semantic concept vector guidance. Existing stochastic vector control can answer whether arbitrary directions are effective, but it cannot answer whether the same semantic direction produces similar authenticity, emotion, rejection, language, or ICL effects without optimizing the forgetting loss on the forgotten set.

- The paper reports numerous results tables but provides little visualization or representational-level analysis to demonstrate how RAd and RAb alter the activation distributions of forgotten, retained, and auxiliary task samples relative to the base model.

- The claimed controllability analysis is insufficiently in-depth. Although the objective function includes scaling factors and the authors tuned the hyperparameters, the paper does not systematically demonstrate how to enhance, weaken, or trade off the interaction between forgetting and memory.

Therefore, I believe that this relatively specific empirical phenomenon is very promising, but the relevant conclusions need to be further refined, better controlled, and more fully demonstrated in terms of mechanism in order to be completely convincing.

**Requested Changes:**

1. Narrow the title and main framework. The current title suggests that forgetting behavior in machine learning often leads to controllable accompanying behaviors and capabilities, but existing evidence primarily focuses on concept-guided representation misdirection in forgetting contexts.

2. Add a guided baseline model using only the same semantic concept vectors. The paper has already compared this to randomized orientation, but this doesn't answer whether the same concept orientation would induce similar accompanying behaviors without forgetting. A useful comparative model should include: the baseline model, the guided model for inference using the same concept vectors and layers, the representation misdirection model using randomized vectors, and the representation misdirection model using semantic concept vectors. The model using only the guided model should be evaluated based on both accompanying task metrics and forgetting metrics. This will help clarify whether the representation misdirection model uniquely links forgetting to accompanying behaviors, or whether accompanying behaviors are simply inherited from the concept vector guided model.

3. Add visualizations or analyses of the representation level. It would be clearer if the paper could show the changes in activation distribution after RAD and RAB. For example, project the forgetting, retention, and auxiliary task activations onto conceptual directions, both before and after forgetting learning; demonstrate whether RAD moves forgotten representations toward the target conceptual region and whether RAB removes the corresponding components; and compare these changes with the base model, stochastic vector RM, and guided-only baseline. This will more directly support the proposed mechanism than simply providing a table of downstream results.

The following improvements will enhance this work:

1. Analyze controllability, not just the occurrence of reported behaviors. The paper should demonstrate how the magnitude of auxiliary behaviors varies with scaling factors, layer selection, or probe quality, and how this variation trades off with the performance of forgetting task suppression and retention tasks.

2. Add a cross-conceptual side effect matrix. For each conceptual direction, report not only the expected target behavior but also other measured behaviors. For example, assess whether the authenticity direction affects rejection, affect, language, and ICL, and whether the rejection direction affects authenticity or overall helpfulness.

---

> ### Author Response · Authors · 2026-07-06
>
> We thank the Reviewer for your time and effort in reviewing our paper. Below, we will keep the response to your request changes (RC) brief and focused. The newly added content in the revision is highlighted in **blue text**.
>
> ### Major Changes:
>
> **RC1.** We acknowledge this overgeneralization and are happy to tone down our claims and change the term "Machine Unlearning" to "Representation Misdirection" in the title. We have updated the title to: *"Beyond Forgetting: Representation Misdirection Elicits Controllable Side Behaviors and Capabilities"*
>
> **RC2**. We thank you for this suggestion. We would discuss that since RAd and RAb are training based methods that are designed with specific objectives (forgetting and retaining) while concept-guided inference-time steering works at inference and has only one objective. We believe that directly comparing concept-guided inference-time steering with our RAd and RAb is conceptually mismatched.
>
> We would emphasize that RAd and RAb are training-based methods specifically designed to optimize both forgetting and retaining objectives. In contrast, concept-guided inference-time steering is an inference-time intervention with a fundamentally different objective,  only steering model behavior at inference without parameter updates. Consequently, we believe that a direct comparison between concept-guided inference-time steering and RAd/RAb is conceptually mismatched. Intuitively, concept-guided inference-time steering can elicit the behaviors associated with the high-level concept, but it fails to produce effective unlearning.
>
> To investigate this, we **conducted additional experiments to investigate whether the RAd/RAb uniquely offer unlearning with side behaviors or whether these behaviors are inherited from the concept vector in inference-time steering**. We implement two inference-time settings:
>
> (1) **concept-guided addition**, which performs inference-time steering by adding the concept vector without parameter updates;
>
> (2) **concept-guided ablation**, which performs inference-time ablative steering using the concept vector.
>
> We evaluate both settings on Zephyr-7B and Mistral-7B while varying the steering coefficient $c$, using TruthfulQA (open-ended and multiple-choice) to measure side behaviors and MMLU and WMDP to assess unlearning performance.
>
> Our results (shown in Figure 2, Section 5.2) reveal that while inference-time steering can elicit side behaviors, it fails to unlearn.
>
> **RC3**. To better understand how RAd and RAb manipulate the model’s latent space, we **conducted an additional experiment in Section 5.1** to analyze the projection of latent representations onto the target concept direction.
>
> ### Additional improvements:
> > 1. Analyze controllability, not just the occurrence of reported behaviors. The paper should demonstrate how the magnitude of auxiliary behaviors varies with scaling factors, layer selection, or probe quality, and how this variation trades off with the performance of forgetting task suppression and retention tasks.
>
> **Answer:** We would clarify that in Appendix G.4, we presented an ablation study about the probe quality, such as (1) how many samples are needed before the induced behaviors become reliable, and (2) comparing logistic regression to alternative probes (e.g., Ridge Regression, K-Means clustering) or simpler contrastive methods (e.g., difference-in- means). An additional experiment on the effects of layer selection is conducted in Appendix G.5.
>
> > 2. Add a cross-conceptual side effect matrix. For each conceptual direction, report not only the expected target behavior but also other measured behaviors. For example, assess whether the authenticity direction affects rejection, affect, language, and ICL, and whether the rejection direction affects authenticity or overall helpfulness.
>
> **Answer:** We thank you for this suggestion. We would clarify that our empirical analysis in Sec. 4.1.4 (Language control) and Sec. 4.2.1 (Linguistic and knowledge tasks) already provides a rigorous cross-evaluation that we believe addresses this concern.
>
> In Tab. 6, we evaluate the Language Presence Rate (LPR) across four different language-specific directions (en $\to$ fr, en $\to$ es, en $\to$ ja, en $\to$ vi). The results show that steering the model with a specific language vector (e.g., en $\to$ fr) significantly increases the presence of French while having a minor effect on the presence of others.
>
> In Tab.7, we conduct a cross-evaluation for ICL across four distinct domains: antonyms, present-to-past, country-to-capital, and person-to-language. The results show that RAd with the ctry->cap direction boosts accuracy on the country-to-capital task from 24.6% to 69.0% (for Zephyr-7B) while having a minimal effect on other tasks.
>
> **Once again, we thank the Reviewer for helping improve our work. We are committed to addressing remaining concerns in the final revision.**